# Winner-Takes-All or Co-Evolution among Platform Ecosystems: A Look at the Competitive and Symbiotic Actions of Complementors

**Yuki Inoue**

Human Augmentation Research Center, National Institute of Advanced Industrial Science and Technology, 2-3-26, Aomi, Koto-Ku, Tokyo 135-0064, Japan; yuki.inoue@aist.go.jp

**Abstract:** Technological platforms such as hardware or systems form platform ecosystems, which are communities orchestrated by platform providers, outside complementors such as software providers, and consumers. Previous studies have suggested that a winner-takes-all situation among platform ecosystems could be induced by interactions between complementors and consumers. However, our observation of the Japanese video game market over the last 30 years indicated that complementors (i.e., software providers) usually seek to avoid winner-takes-all situations and, instead, promote symbiotic situations. Using the Lotka–Volterra equations from biology as a reference, we developed a model to understand the competitive behavior of complementors among platform ecosystems. We used a 19-year (1996–2015) dataset on the Japanese video game market and confirmed that complementors took as many actions to create symbiotic situations as they took to create winner-takes-all situations, if not more. Our results show that such actions by complementors are influenced by several factors of platform ecosystems. This study also suggests that certain complementors that contribute to symbiotic co-existence within a platform ecosystem could emerge as keystone firms/companies. These complementors could contribute to the sustainability of platform-based markets and facilitate the co-existence of multiple platform ecosystems.

**Keywords:** platform ecosystem; two-sided market; winner-takes-all; competitive Lotka–Volterra equation; video game market; competition/symbiosis; open innovation

## 1. Introduction

### 1.1. Platform ecosystems

Information society has given rise to platform-based markets. These platforms include not only hardware platforms such as video game consoles or personal computers, but also intermediation platforms on the web. Research on traditional platforms has investigated from two perspectives [1,2]. The first focuses on the "product platform." By sharing compatible systems and architecture on a product platform, the platform promotes efficient product development and ultimately facilitates innovation. The second focuses on the "intermediary platform." An intermediary platform creates a multi-sided market (or two-sided market if there are two groups, such as buyers and sellers) by acting as an intermediary among participants from multi-sided groups. These have recently been integrated to establish the "platform ecosystem" type [1–3]. Therefore, researchers of platform ecosystems focus on two functions: the compatibility function and the intermediary function.

Platform ecosystems are communities orchestrated by platform providers, outside complementors, and consumers. An ecosystem in the business context is referred to as a "business ecosystem." A business ecosystem is an "economic community supported by a foundation of interacting organizations

and individuals—the organisms of the business world; the member organisms also include suppliers, lead producers, competitors, and other stakeholders" [4,5]. A platform ecosystem restricts its scope to actors related to the platform, such as the platform and its providers and users. However, platform ecosystems do not generally restrict participation in or withdrawal from them. Therefore, although a platform ecosystem has boundaries, it is an open system. Accordingly, the composition of a platform ecosystem may not necessarily converge to any specific states owing to interaction with the outside.

A platform ecosystem is made up of the platform, as a system or an architecture, and a collection of supporting complementary assets [2,6,7]. Among complementary asset providers, those that produce complementary goods for the platform are called "complementors" [8]. A platform ecosystem includes three kinds of actors: platform providers, complementors, and consumers. Platform providers provide their own platforms. Complementors develop and/or provide complementary goods (products and/or services) using platform technology. Consumers purchase the complementary goods provided by complementors via the platform. A platform ecosystem can foster unlimited innovation via the participation of various organizations that possess several management resources as complementors [1]. It also induces consumers with varying needs to adopt the platform [9]. The success of a platform ecosystem depends upon the success of the entire ecosystem [10]. Even an innovative and technologically superior platform ecosystem cannot be sustained if the complementors related to the development and provision of goods are not successful. Nintendo Wii is an example of a failure resulting from the complementors' failure to use the platform technology, even though the platform became widely used [3].

The main focus of platform ecosystem research is information technology (IT) platforms. The video game market is a representative platform ecosystem: hardware is the platform, software is the complementary good, software providers are complementors [11–14]. Platforms in the video game market (video game hardware) function as both product platforms and intermediary platforms. First, the platforms help external software developers and providers to create products more efficiently and effectively by reducing the cost of hardware development and providing technologically superior platforms. Second, since the developed and provided software is compatible with the platform, consumers introduced to the platform can play it. In addition, software providers can reach the consumers who have been introduced to the platform (installed base). Other examples of such relationships include those between operating systems and application software [15–17], system platforms and enterprise software [9], and web browsers and extensions [18]. Studies have examined the development of and competition among platform ecosystems [11,12,15], the growth mechanism of complementors [9,16], the diversity of the complementary goods of complementors [17], and development and competition within the platform ecosystem [18].

Platform ecosystems represent one type of open innovation. Two types of open innovation pattern can occur with a platform ecosystem. First, public innovation occurs if the platform itself is considered the innovation and the complementors simply use it [19]. Open-source innovation occurs if the platform and complementary goods realize the innovation jointly [19]. A platform ecosystem is an open business model [20]. In this model, platform providers can indirectly use complementors' resources for the platform as complementary goods, thereby increasing the value of the platform. Additionally, firm's technology openness strategy, complex adaptive systems, and market responses stimulated by technology innovations are important for successful open innovation [21]. While these factors are significant for the success of platform ecosystems, platform providers must also manage the environments in which complementors use the platform technology to develop their complementary goods [3].

*1.2. Winner-Takes-All Competitive Mindset*

The winner-takes-all mindset may be induced by the influence of indirect network effects whereby, in a two-sided market, as the scale grows on one side, profits increase on the other [22–26]. This effect implies that complementors and consumers interact and that the number of complementors and

customers grows exponentially; it could thus produce a winner-takes-all market in which a single platform takes almost all of the complementors and consumers [27–29]. The possible influence of direct network effects would accelerate the winner-takes-all situation [28,30].

Researchers have suggested that winning in a winner-takes-all market by exerting indirect network effects requires first establishing an installed base [11,12,31]. A platform with a smaller installed base will face negative growth via the indirect network effects mechanism if the platform lacks specialized markets [28]. Since superior complementary goods promote the total sales of the platform [32], the platform providers themselves often offer attractive complementary goods [13,33] or induce capable complementors to do so [34,35]. However, even if a platform achieves a large installed base, the lack of a sufficient complementary goods volume would cause the platform ecosystem to decline [36]. Accordingly, platform providers should pay attention to the profits not only of consumers but also of complementors.

*1.3. Focus and Purpose of Study*

Common complementors can be shared among multiple business ecosystems formed by multiple core companies [37]. This is also applicable to platform ecosystems; multiple platform ecosystems can share complementors. Since platform ecosystems comprise two-sided markets between complementors and consumers, platform ecosystems can compete to acquire complementors, leading to winner-takes-all situations.

However, we can observe exceptions. For example, Figure 1 shows the concentration of new goods provision (game software) for stationary-type hardware platforms in the Japanese video game market. The y-axis shows the Herfindahl–Hirschman Index (HHI) values of the new game software provision for each platform for each fiscal year (April 1 to March 31). HHI is a representative indicator that calculates competition between companies in a market. If we assume the market share of each company $x$ to be $S_x$, $S_x^2$ becomes HHI. Here, to measure the degree of competition in the provision of game software for each platform, we calculated HHI based on the share of the number of new game software units in each year. The figure also describes the starting points for each generation of the hardware platforms. Table 1 lists the platforms included in the calculation of Figure 1. The figure shows that the degree of the market concentration of software provision (in terms of market share) tends to decrease a few years after the release of each of the hardware generations. This suggests that complementors in this market tend not to concentrate on providing all goods to one platform that may have acquired a large installed base. Conversely, these complementors may also have acted to avoid a winner-takes-all situation in their platform ecosystems. We can also observe an alternation between moves for winner-takes-all and moves for symbiosis. Here, we define symbiosis as the converse of a winner-takes-all situation, where multiple platforms co-exist stably.

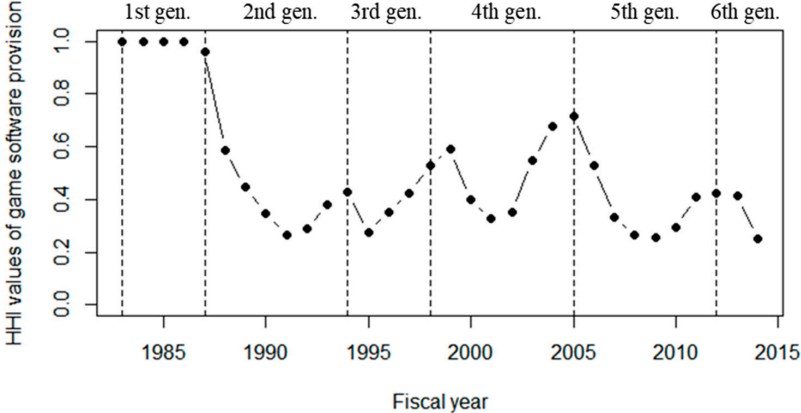

**Figure 1.** Market concentration of new good provision in the Japanese video game sector. Data are taken from Japanese video game database f-ism.net (http://www.f-ism.net/, accessed on 11 December 2018).

**Table 1.** Stationary-type hardware platforms in Japanese video game sector

| Generation | Platform Name |
| --- | --- |
| 1st (standard for following generations) | Nintendo Entertainment System, Sega Master System. |
| 2nd | PC Engine, MEGA DRIVE, Super Nintendo Entertainment System, NEOGEO. |
| 3rd | NEOGEO CD, SEGA SATURN, PlayStation, PC-FX, VIRTUAL BOY, 3DO, NINTENDO64. |
| 4th | Dreamcast, PlayStation 2, NINTENDO GAMECUBE, Xbox. |
| 5th | Xbox 360, PlayStation 3, Wii. |
| 6th | Wii U, PlayStation 4, Xbox One. |

Some scholars argue that a winner-takes-all situation does not always occur. Even if a platform is a late mover and has a small installed base, technological superiority and the expectation of a future provision of complementary goods could enable the development of the platform ecosystem [12]. Although technological platform superiority is not essential for the creation of an ecosystem, it can be a factor in ecosystem creation. Whether the technological superiority of a platform contributes to the creation of an ecosystem depends more on the degree to which consumers can obtain utility from such technical superiority than on other factors such as indirect network effects. The survival of a late mover could also be aided by the provision of new complementary goods that reduced the value of old goods by making them obsolete [38]. If the first mover makes a positioning error, the late mover will a chance to grow if it avoids making the same error [28]. Furthermore, when the positioning of the platform that depends on complementary goods provision is differentiated from the competitive platforms, the degree of the winner-takes-all situation decreases [14]. In technological terms, a platform structure that allows an easy providing of the same complementary goods across multiple platforms can decrease the occurrence of winner-takes-all situations [39,40]. Thus, the degree of winner-takes-all competition can be influenced by the expectations for the platform, the distinctiveness of the platform ecosystem, and the ease of multi-participation among the platforms.

However, studies have focused on specific cases and factors. As shown in Figure 1, we can consider the possibility that complementors take action to avoid winner-takes-all situations and induce symbiotic situations more frequently. Studies have also failed to explain the alternation between winner-takes-all and symbiotic situations, as shown in Figure 1. Whether the relationships among platform ecosystems result in a winner-takes-all or symbiosis situation is a critical question for the survival not only of the platform providers but of all market participants. Accordingly, this study poses two research questions:

*Are complementors' symbiotic actions as frequent as their winner-takes-all actions, and what factors influence their competitive and symbiotic activities?*

This study has two purposes: (a) to confirm that complementors usually act to avoid a winner-takes-all situation (i.e., to induce a symbiotic situation), and (b) to comprehensively clarify the underlying mechanisms of complementors' behavior which brings winner-takes-all (or symbiotic) situations.

This study should provide new insights into the sustainability of platform ecosystems and platform-based markets. Failing to embrace sustainability in a winner-takes-all single platform ecosystem situation is risky. If competition produces a winner-takes-all scenario, the market will be monopolized by a single platform and provider. In such a situation, the platform provider has a free hand to maximize profits, which would demand more profit allocation from both the complementors and the consumers. This would cause the complementors and consumers to become unsustainable within the ecosystem and in turn reduce the stability and survivability of the platform ecosystem itself. Thus, understanding the symbiotic mechanisms used to avoid winner-takes-all situations

among platform ecosystems will help improve ecosystem environments and structures and thus enhance sustainability.

The term "symbiosis" is not as common as "competition" in the field of business management and has no uniform definition. The term is used in environmental management research [41] to refer to the "co-existence of the environment and industry" and in business ecosystem research [42,43] to refer to "cooperation and co-existence." The latter definition is close to that used in this study. [42] suggested that the symbiosis of a business ecosystem shares the fate of the network as a whole, irrespective of the apparent strength of the network members. Valkokari [43] indicated that an ecosystem is composed of providers and consumers who benefit from the interaction and are thereby intertwined in relationships that can be symbiotic. Thus, some of the research related to business ecosystems focus on symbiosis in the ecosystem. However, no studies have analyzed competition and symbiosis simultaneously in the business market. This study analyzes symbiosis in a manner equivalent to "competition" in the context of a business and platform ecosystem.

### 1.4. Definition of Influence α Related to Winner-Takes-All Situations for Complementors

To evaluate the actions of complementors that lead to winner-takes-all situations, we considered how complementors participating in a platform are influenced by the performance of competing platform ecosystems. Let us define "influence $\alpha$" as the degree to which complementors are affected by the performance of competing platform ecosystems. We can then depict the actions of complementors that lead to winner-takes-all situations as shown in Figure 2. Complementors participating in a platform who are negatively influenced by the performance (degree of growth/decline) of competitor platform ecosystems adopt a winner-takes-all strategy. Conversely, complementors who are positively influenced by the performance of competitor platform ecosystems avoid adopting a winner-takes-all strategy, and the platform grows (or declines) symbiotically along with the other platforms. Complementors that are not influenced by other platform ecosystems decide to invest in the platform independently of other platforms. Even if there is only one market, we can consider situations in which platforms are regarded as independent of each other. Specifically, we suppose that an independent situation could occur when complementors consider that the evolution of platforms A and B do not depend on the evolution of another platform or on each other. As an extreme example, considering that most of the game software on platform A is for children, and most of the game software on platform B is for adults, we propose that complementors would consider these platforms to be independent of each other.

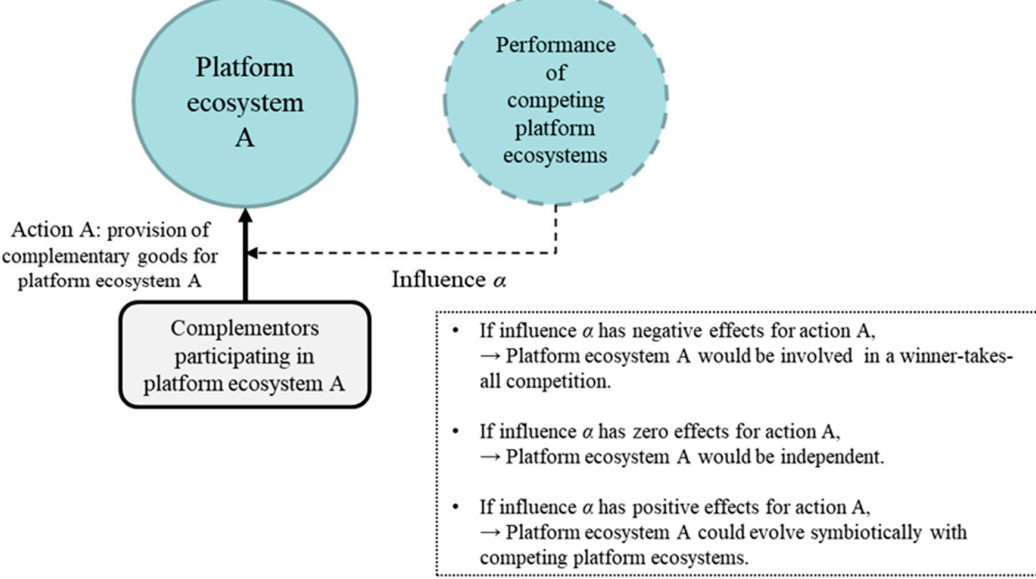

**Figure 2.** Description of influence α and actions of complementors for the occurrence of a winner-takes-all situation

*1.5. Possible Influencial Factors on Complementors' Competitive or Symbiotic Action*

We considered the factors that could change the value of influence α. Previous studies have not investigated the changes between competition and symbiosis in platform ecosystems. However, several studies discuss the factors that influence or are related to complementors' behavior. These address not only the complementors in platform ecosystems but also the firms providing products. This study examined the factors that may affect such complementors' behavior.

First, we considered the factors related to the relationships among platform ecosystems. As the number of consumers using a platform increases, the participation of the complementors on that platform will also increase due to indirect network effects [11,12]. Amid competition among platforms, as a platform gains market share, the complementors operating on it will also increase [44]. Accordingly, we assume that the higher the degree of market monopolization, the more complementors move among platforms, and the more competitive the relationships among platform ecosystems become. We therefore assume that the degree of monopolization could influence α.

For video game software, complementary goods are in the software product category. When the positioning of a platform based on software is differentiated, winner-takes-all intensity lessens [14]. Conversely, when there is a high similarity between software genres among platforms, competition among them increases. Thus, we assume that software genre similarity among platforms can influence the value of α.

Complementors need to make decisions about the allocation of goods provision on the platform. A situation in which a consumer or complementor joins multiple platforms is known as "multi-homing" [23,45]. Among multi-homing types, cross-platforms, which provide the same complementary goods to multiple platforms, have the advantage of being able to disperse the goods development cost for consumers over several platforms [39]. In addition, a platform architecture that facilitates cross-platforms decreases the possibility of winner-takes-all situations [40]. The degree to which such platform relationships exist among multi-homing types is reflected in the degree of platform embeddedness. Thus, we assume that the degree of complementors' embeddedness in a platform could influence the value of α.

Video game platforms tend to alternate generations every few years or decades. In this process, the compatibility of related platforms can be secured, or the same software can be provided to them. Since the software provision of complementors might be influenced by this alternation process, the process could affect the value of α.

As mentioned, we considered four factors associated with the relationships among platform ecosystems that may influence the actions of complementors. Accordingly, this study proposes the following:

**Hypothesis 1.** *The value of influence α (the degree to which complementors are affected by the performance of competing platform ecosystems) is affected by four factors related to the relationships among platform ecosystems: Degree of monopolization of platforms in the market, Similarity of product category of complementary goods, Embeddedness of complementors in the platforms, and Influence of related platforms.*

Second, we considered the factors related to the environment of a platform ecosystem. Even if a platform acquires an adequate installed base and achieves high sales volume, complementors will change their participating platforms if they cannot earn more profits than they earn on competing platforms [3,46]. Accordingly, the degree of complementors' sales volume bias could influence the value of α.

Bias (or uniformity) in the product category of complementary goods influences complementors' product development. For example, when the scale of the product category of complementary goods is biased, complementors could find it difficult to establish their positioning [47] and may be tempted to imitate competitors [48]. Thus, since bias (or uniformity) in the sales of the product category of complementary goods could influence the goods' provision, it could also influence the value of α.

Platform ecosystems grow and decline not only because of the technological product lifecycle but also because of the lifecycle of the ecosystems themselves. Since video game hardware is a technological system, a hardware platform loses its superiority or edge when new technology or hardware is introduced. However, since the adoption of a platform is dependent on both complementors and consumers, the ecosystem's growth and decline do not always depend on the technological superiority of the platform [3,46]. In addition, the state of a platform ecosystem reflects the actions of complementors. Accordingly, the growth and decline of an ecosystem could influence complementors' decision to pursue an imitation [48] or bandwagon effects [49–51]. Thus, the degree of growth or decline in a platform ecosystem could also influence the value of $\alpha$.

We considered three factors related to the environment of a platform ecosystem that might influence the actions of complementors. Accordingly, this study proposes the following:

**Hypothesis 2.** *The value of influence $\alpha$ (the degree to which complementors are affected by the performance of competing platform ecosystems) is affected by three factors related to the platform ecosystem environment: Bias of sales volume of complementors, Bias of scale of the product category of complementary goods, and Growth and decline of the platform ecosystem.*

Finally, we considered the factors related to the culture of the complementors in the platform ecosystem. Companies form a dynamic capability by coevolving learning mechanisms, such as by experience accumulation, knowledge articulation, and knowledge codification processes [52]. This organizational learning influences innovation and performance [53]. The quality of complementary goods influences the strength of the indirect network effects on consumers' platform adoption [54] and is deemed by consumers to reflect the quality of the platform [55]. Thus, platform providers need complementors that have gained experience drawn from organizational learning in the market and that can develop superior complementary goods. Conversely, complementors with considerable experience in the market might change how they engage with the platform ecosystem. For example, since they have accumulated the know-how required to develop complementary goods, they could find multi-homing on multiple platforms easier. Alternatively, they may understand the technical characteristics of each platform and focus on a specific platform. Thus, experience in the market could also influence the value of $\alpha$.

Complementors pay sunk costs to participate on a platform [56], such as an introduction fee to develop the environment and the effort expended to reorganize it. According to the sunk cost fallacy [57], these sunk costs could force complementors to stay on a platform despite falling profitability. Thus, the degree of new participation on a platform could also influence the value of $\alpha$.

As mentioned above, we considered two factors related to the culture of complementors in the platform ecosystem that might influence complementors' actions. Accordingly, this study proposes the following:

**Hypothesis 3.** *The value of influence $\alpha$ (the degree to which complementors are affected by the performance of competing platform ecosystems) is affected by two factors related to the culture of complementors in the platform ecosystem: Degree of experience in the market and Degree of new participation on the platform.*

*1.6. Framework of Analysis*

This study consists of three steps. First, we use competitive Lotka–Volterra equations drawn from biology as a reference, adapt them to a platform-based market, and develop our model for influence $\alpha$. Second, we calculate the value of influence $\alpha$ using a dataset comprised of video game markets (analysis 1). Third, we test how influence $\alpha$ is affected by the relationship among platform ecosystems, environment, and complementor culture (analysis 2). This framework is depicted in Figure 3.

| |
|---|
| **- Modeling -** <br> Construction of the model expressing influence $\alpha$ using the competitive Lotka–Volterra equations from biology. |

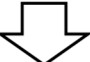

| |
|---|
| **- Analysis 1 -** <br> Calculation of the value of influence $\alpha$ using the dataset of video game markets. |

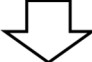

| |
|---|
| **- Analysis 2 -** <br> Analysis of how influence $\alpha$ is affected by the relationship among platform ecosystems, the environment, and culture of complementors. |

**Figure 3.** Framework of analysis

While competitive Lotka–Volterra equations are mainly used in biology, several management studies have also used them. For example, Morris and Pratt [58] theorized about the technological substitution between existing dominant systems and new, competing systems. Watanabe, Kondo and Nagamatsu [59] analyzed the predator–prey relationship between new and existing technologies. López-Sánchez, Arroyo-Barrigüete and Ribeiro [60] developed a model of competition among competing products. Michalakelis, Christodoulos, Varoutas and Sphicopoulos [61] focused on the development of the high-technology market, which is saturated with dominant players. Tseng, Liu and Wu [62] predicted scenarios for smartphone OS by using Lotka–Volterra equations and Delphi methods. Thus, researchers have used the competitive Lotka–Volterra equations to analyze the development and competition of systems or technologies.

## 2. Materials and Methods

### 2.1. Modelling

In this subsection, we elaborate on our analytical model that expresses influence $\alpha$.

2.1.1. Competitive Lotka–Volterra Equations

In biology, competitive Lotka–Volterra equations contain, on the left-hand side, a variation in the number of a species $x$ as $dN_x/dt$ at period $t$ and, on the right-hand side, the number of a species $x$ as $N_x$, the carrying capacity for species $x$ as $K_x$, the intrinsic rate of growth of species $x$ as $r_x$, the number of other species $y$ as $N_y$, and coefficients. When $n$ species exist, the equation is written as

$$\frac{dN_1}{dt} = \frac{r_1 N_1}{K_1}(K_1 - N_1 - \alpha_{1,2}N_2 \ldots - \alpha_{1,n}N_n), \tag{1}$$

$$\vdots$$

$$\frac{dN_n}{dt} = \frac{r_n N_n}{K_n}(K_n - \alpha_{n,1}N_1 - \alpha_{n,2}N_2 \ldots - N_n). \tag{2}$$

where $\alpha_{x,y}$ is the coefficient of competition of the $y$th species on the $x$th species [63].

When $\alpha_{A,B}$ has a positive value, species B is in competition with species A. In this scenario, the growth of species B leads to a decline in species A, since it decreases its carrying capacity. Conversely, when $\alpha_{A,B}$ has a negative value, species B is symbiotic with species A. In this case,

the growth of species B leads to a growth in species A, since it increases its carrying capacity. When $\alpha_{A,B}$ is zero, species B is neutral with species A. In this case, the growth of species B does not influence the growth of species A. If we consider the relationship between the two species, for each $\alpha$, "$- -$" represents mutualism, "$+ +$" represents competition, "$+ -$" or "$- +$" represents predation or parasitism, respectively, "$- 0$" or "$0 -$" represents commensalism, "$+ 0$" or "$0 +$" represents amensalism, and "$0\ 0$" represents neutralism [64].

If we adapt Equation (2) for competition among platform ecosystems, we define $n$ as a type of platform (in this study, $n$ corresponds to each video game hardware), $N$ as the scale of the complementary goods provided (in this study, $N$ corresponds to the number of provided video game software units), $K$ as the capacity for consumer purchases of complementary goods (in this study, $K$ corresponds to the capacity for consumers' purchases of video game software, and the value is standardized as a scale of $N$), $\alpha$ as the influence of the complementary goods on competing platforms for consumer purchases on platform $n$, and $r$ as the rate of change of complementary goods provision (in this study, $r$ corresponds to coefficients related to the degree of change in the provision of video game software). Details on each measure are provided in Sections 2.2.2.1 and 2.2.2.2. The coefficient of competition $\alpha$ corresponds to influence $\alpha$ (defined in Section 1.4), by which the performance of competing platforms influences the complementor goods provision on a platform. However, it must be noted that an increase in the value of influence $\alpha$ could signify a winner-takes-all situation in the Lotka–Volterra equations.

This study defines "carrying capacity" as the amount of complementary goods that can be provided in the platform ecosystem. The term "installed base" has been used as an equivalent indicator in the platform research [11,12]. "Installed base" refers to the scale of the consumers who have introduced the platform. The scale of the installed base can be regarded as an indicator of the degree of total demand for complementary goods on that platform [15]. Therefore, complementors can refer to the scale of the installed base to make decisions about the provision of complementary goods. However, platforms become obsolete or lose the interest of consumers. Therefore, the capacity for complementary goods, as represented by the installed base, changes over time. Accordingly, researchers who have used indicators related to the installed base have multiplied the installed base by a decay coefficient [12], multiplied it by elapsed time to reproduce the inversed-U pattern [11], or used consumer purchase size as the indicator for complementary goods capacity [3]. Since this study developed models for empirical analysis based on the Lotka–Volterra equation, we used the term "carrying capacity" as it is.

In Equation (2), we assume that complementors provide their goods until the amount of goods provision and the scale of consumer purchases reach equilibrium. This assumption corresponds to the relationship between the amount of the installed base and that of complementary goods provision [15]. Figure 4 depicts this concept. In Equation (2), influence $\alpha$ is expressed as the degree by which the volume of complementary goods on other platforms affects the capacity expectations about consumer purchases on this platform. Figure 5 depicts these expressions.

### 2.1.2. Differences between Competition among Species and among Platform Ecosystems

Despite adopting competitive Lotka–Volterra equations to express competition among platform ecosystems, we are aware that competition among species and competition among platform ecosystems have different mechanisms. Accordingly, we specify those differences to modify the model.

First, whether the carrying capacities are actual values or not (i.e., expected values) is an important issue. For species, the scale of carrying capacity influences the scale of individuals in real time. However, for platform ecosystems, the scale of carrying capacity (volume of consumer purchases of complementary goods) cannot influence the scale of individuals (provision of complementary goods) in real time, since the date of the goods' release and the date when the goods provision is decided are different.

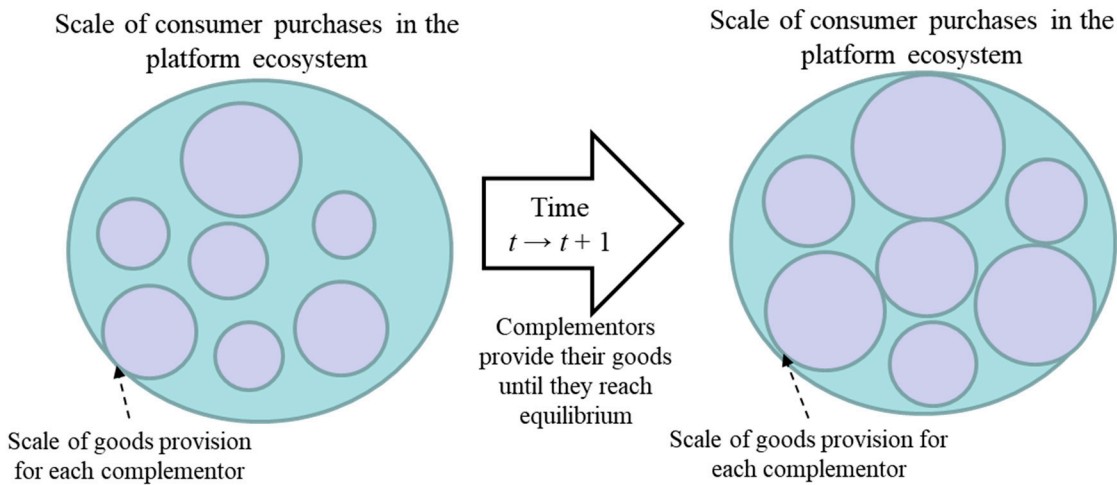

**Figure 4.** Expressing complementor goods provision in competitive Lotka–Volterra equations

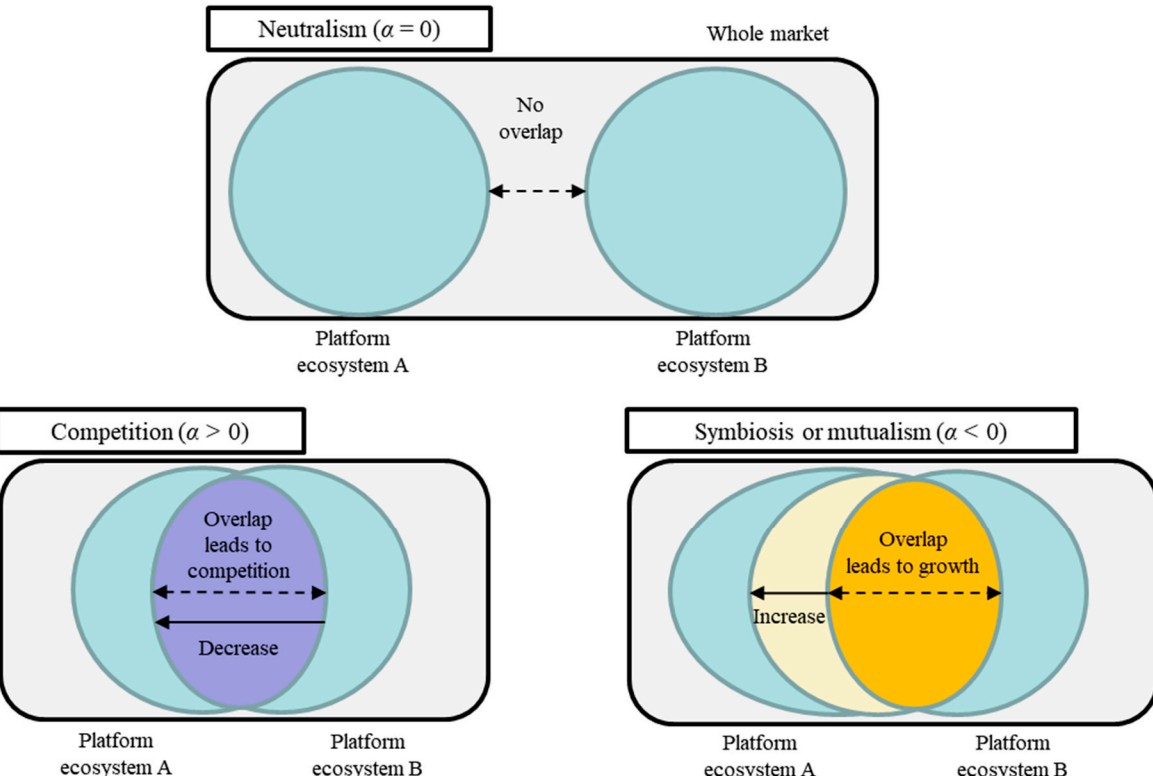

**Figure 5.** Expression of influence $\alpha$ in competitive Lotka–Volterra equations

Second, the entry of participants from outside is another important issue. Outside participants can join the platform ecosystem (e.g., movers from other platforms or new entrants). Since competitive Lotka–Volterra equations are based on variations in participating individuals, we need to exclude data on complementors from outside in our empirical analyses for each time period *t*.

Third, the heterogeneity of individuals is an important issue. The performance of an individual complementor (i.e., sales volume) could be more biased than that of a species (i.e., amount to eat). Accordingly, we need to adjust for the influence of heterogeneity in our empirical analyses.

*2.2. Empirical Analysis*

In this subsection, we explain the methods used in the empirical analysis. First, we explain our dataset. Second, we explain the methods used for analysis 1 (estimating the value of influence $\alpha$). Third, we explain the methods used for analysis 2 (mechanisms of change in the value of influence $\alpha$).

2.2.1. Dataset

The video game industry is a representative example of a platform-based market and is also a significant contributor to the global entertainment economy [65]. In the video game industry, a two-sided market has emerged between software providers and consumers. Their interaction creates indirect network effects [11,12]. Many studies on platform ecosystems have analyzed the video game market [3,11–14,46]. Therefore, we consider that the video game market is an appropriate subject for an analysis of platform ecosystems.

This study used data drawn from the Japanese video game database f-ism.net (http://www.f-ism.net/, accessed on 11 December 2018), maintained by the Kadokawa Dwango Corporation. From this database, we obtained the title name, provider platform, supplier, price, release date, monthly sales, and genre for each game software. The data cover April 1996 to March 2015. We acquired a dataset comprised of 15 hardware platforms, as shown in Table 2. We compiled these and calculated the variables for the number of new software units, the quantity of software sales, and the proportion of each genre in each month for each platform. We removed samples for which the number of new software units or software sales volume was 0. The sample comprised 1240 observations, an adequate sample size relative to the samples used by previous studies on video game markets (e.g., [3,11,12,14,46]). This study used data only from the Japanese market to avoid inconsistencies, since the preferences of game players strongly depend on regional and cultural factors [66]. The detailed procedures used to formulate the analytical model are enumerated below (Section 2.2.2.2). Since the data are provided in a time series, an autocorrelation problem in the statistical analysis is a possibility. To address this possibility, we conducted the appropriate treatment, as described in Section 2.2.3.4.

The dataset required preprocessing. First, since the original 22 genres were too many for the analysis, we integrated small genre samples with similar genres. In addition, we labeled the different types of software (i.e., standard, resell at low price, bundled, and limited version) based on the software names. Since the conception and development processes involved in providing these special types of software differ from those involved in standard software, we removed the special types of software from the calculation of the indicators related to software provision.

2.2.2. Analysis 1: Estimating the Value of Influence $\alpha$

2.2.2.1. Modification of Competitive Lotka–Volterra Equations

The purpose of analysis 1 is to estimate and extract the value of influence $\alpha$. We assumed that the value varies over time since the market conditions of the platform ecosystem are not static. Accordingly, we modified the competitive Lotka–Volterra equations to adapt them to the model for the platform ecosystems.

**Table 2.** Platforms used in the analysis of influence $\alpha$

| Platform Name | Preceding Platform | Succeeding Platform | Provider | Type | Release Date in Japan | Gene-Ration |
|---|---|---|---|---|---|---|
| SEGA SATURN | MEGA DRIVE | Dreamcast | SEGA | Stationary | Nov. 1994 | 3 |
| PlayStation | - | PlayStation 2 | Sony Computer Entertainment | Stationary | Dec. 1994 | 3 |
| PC-FX | PC Engine | - | NEC | Stationary | Dec. 1994 | 3 |
| NINTENDO 64 | Super Nintendo Entertainment System | NINTENDO GAMECUBE | Nintendo | Stationary | Jun. 1996 | 3 |
| Dreamcast | SEGA SATURN | - | SEGA | Stationary | Nov. 1998 | 4 |
| PlayStation 2 | PlayStation | PlayStation 3 | Sony Computer Entertainment | Stationary | Mar. 2000 | 4 |
| NINTENDO GAMECUBE | NINTENDO 64 | Wii | Nintendo | Stationary | Sep. 2001 | 4 |
| Xbox | - | Xbox 360 | Microsoft | Stationary | Feb. 2002 | 4 |
| Nintendo DS | GAMEBOY ADVANCE | Nintendo 3DS | Nintendo | Portable | Dec. 2004 | 5 |
| PlayStation Portable | - | PlayStation Vita | Sony Computer Entertainment | Portable | Dec. 2004 | 5 |
| Xbox 360 | Xbox | Xbox One | Microsoft | Stationary | Dec. 2005 | 5 |
| PlayStation 3 | PlayStation 2 | PlayStation 4 | Sony Computer Entertainment | Stationary | Nov. 2006 | 5 |
| Wii | NINTENDO GAMECUBE | Wii U | Nintendo | Stationary | Dec. 2006 | 5 |
| Nintendo 3DS | Nintendo DS | - | Nintendo | Portable | Feb. 2011 | 6 |
| PlayStation Vita | PlayStation Portable | - | Sony Computer Entertainment | Portable | Dec. 2011 | 6 |

We focused on the variations in the number of game software provided on platform $f$ during time period $t$. This study set the mean time lag between the decision to provide software and the release date as one year (12 months). The variation in the number of game software provided, $\frac{dN_{f,t+12}}{dt}$, is described as

$$\frac{dN_{f,t+12}}{dt} = \frac{r_f N_{f,t}}{K_{f,t}}\left(K_{f,t} - N_{f,t} - \alpha_f N_{g,t}\right) + \varepsilon_{f,t}, \tag{3}$$

where $g$ is the group of platforms competing with platform $f$.

To calculate the time variation in influence $\alpha$, we set moving time windows and calculated the influence $\alpha$ for each one. The time window $s$ is set as 12 months, and $s = t - 11, \ldots, t$. Next, Equation (3) is modified to

$$\frac{dN_{f,s+12}}{dt} = \frac{r_{f,t} N_{f,s}}{K_{f,s}}\left(K_{f,s} - N_{f,s} - \alpha_{f,t} N_{g,s}\right) + \varepsilon_{f,s}. \tag{4}$$

We estimated the optimal value of influence $\alpha_{f,t}$ for each time period $t$ on platform $f$ using Equation (4).

### 2.2.2.2. Application of Dataset

We plugged the values from our dataset into Equation (4). $\frac{dN_{f,s+12}}{dt}$ is given as the variation in the number of game software provided on platform $f$ in time period $s + 12$ by complementors already participating in platform $f$ in time period $s$. Next, the number of game software provided in time period $s$ on platform $f$ gave us $N_{f,s}$. $K_{f,s}$ is given as the indicators for the total software sales volume on platform $f$ in time period $s$.

However, we needed to modify the scale of software sales volume to reflect the scale of the game software provided. Accordingly, we set the total software sales volume as $Y_{f,s}$, standardized from $Y_{f,s}$ to $K_{f,s}$ as $K_{f,s} = CY_{f,s}^{\varphi}$. $C$ and $\varphi$ are calculated by the following equation:

$$\ln\left(N_{f,s}\right) = \ln(C) + \varphi \ln\left(Y_{f,s}\right) + d_f + \varepsilon_{f,s}, \tag{5}$$

where $d_f$ is a dummy variable for the fixed effect depending on each platform and $\varepsilon_{f,s}$ is an error term. Before the main analysis, we calculated $C$, $\varphi$, and $d_f$ as estimated values using an ordinary least square (OLS) regression from our dataset. In the calculation, we took the moving average of each variable to remove the influence of seasonality.

The procedure for calculating influence $\alpha_{f,t}$ was as follows.

[a]　We assumed a possible range of influence $\alpha_{f,t}$ from $-1.5$ to $1.5$ in units of 0.1. For each time period $t$, we plugged each value of influence $\alpha_{f,t}$ ($-1.5 \sim 1.5$) into Equation (4) and calculated the OLS regression. Here, $r_{f,t}$ is given as the regression coefficients.

[b]　In each platform $f$ and time period $t$, we selected $\alpha_{f,t}$ whose value of adjusted $R^2$ is at its maximum when the value of $r_{f,t}$ is positive.

[c]　To ensure the validity of the estimation of $\alpha_{f,t}$, if the significance of the estimated model on platform $f$ in time period $t$ did not satisfy the condition $p < 0.01$, $\alpha_{f,t}$ was set as the missing value.

[d]　We assumed that decision making for software provision on platform $f$ is more affected by the conditions of platform $f$ than by those of other platforms. Next, when the condition $\left|\alpha_{f,t}\right| > 1$ was satisfied, $\alpha_{f,t}$ was set as the missing value.

In the calculation, the group of competing platforms $g$ included those that are of the same type and generation as platform $f$ (see Table 2). In addition, when the conditions $K_{f,s} = 0$, $N_{f,s} = 0$ or $N_{g,s} = 0$ were satisfied, the data point was removed from the calculation.

From Equation (4), $\left| \alpha_{f,t} \right|$ becomes larger as $N_{f,s}$ become larger than $N_{g,s}$, and vice versa. To remove this bias, we modified the value of $\alpha_{f,t}$ as follows.

[a]　In units of 0.1 of the share of the game software provided ($share = N_{f,s} / \left\{ N_{f,s} + N_{g,s} \right\}$), we calculated the standard deviation of influence $\alpha$ as $\sigma^{\alpha}_{share}$.

[b]　We modeled for the estimation of $\sigma^{\alpha}_{share}$ depending on *share* as $\sigma^{\alpha}_{share} = \beta^{\sigma} share + \beta^{C}$. Then, we estimated $\beta^{\sigma}$ and $\beta^{C}$ using an OLS regression.

[c]　Since the value of *share* ranged from 0 to 1, we set *share* = 0.5 as the standard. Then, we modified $\alpha_{f,t}$ as $\alpha_{f,t} \leftarrow \alpha_{f,t} \left( \sigma^{\alpha}_{0.5} / \sigma^{\alpha}_{share} \right)$.

### 2.2.3. Analysis 2: Mechanisms for Change in Influence $\alpha$

In analysis 2, we tested how the influence $\alpha_{f,t}$ that was calculated in analysis 1, was affected by the relationship among platform ecosystems, the environment, and the culture of participants in the ecosystem as discussed in the subsection of hypothesis building. The following subsections explain the quantification method of each factor.

#### 2.2.3.1. Variables: Relationship among Platform Ecosystems

- Degree of platform monopolization in the market

In the real world, an installed platform cannot be used forever because of breakdowns, end-of life cycles, and consumer boredom. Accordingly, previous studies have considered balancing the scale of goods provision and installed base by adding a decay rate to the installed base [12] or by including the terms of time [11]. This study used software sales to calculate market share, not hardware sales. We described the degree of monopolization of/concentration in the market as $v^{1,MarketHHI}_{f,s}$, and the value was calculated using the HHI of each software sales for platforms, including platform $f$ and competing platforms $g$, in time period $s$.

- Similarity of product category of complementary goods

We described the similarity between software genres as $v^{1,Sim.Comp.}_{f,s}$. It was calculated as follows. [a] We calculated the game software provided for each genre on platform $f$ in time period $s$. [b] We calculated the game software provided for each genre on competitive platforms $g$ in time period $s$. [c] We calculated the Pearson correlation coefficient between the value of [a] and the value of [b]. The correlation coefficient was regarded as $v^{1,Sim.Comp.}_{f,s}$.

- Embeddedness of complementors in the platforms

We let the degree of embeddedness of complementors in the platform be described as $v^{1,Embeddedness}_{f,s}$. We referred to [67] and calculated the value as follows. [a] We set the rate of software provided by complementor $k$ for platform $j$ in time period $s$ as $P_{k,j,s}$ (i.e., $P_{k,j,s} = N_{k,j,s} / N_{k,s}$). [b] The degree of embeddedness of complementor $k$ for a single platform is described as $e_{s,k} = \sum_{j} \left( P_{k,j,s} \right)^2$.

[c] We calculated $v^{1,Embeddedness}_{f,s}$ as: $v^{1,Embeddedness}_{f,s} = \left( \sum_{i}^{N_{f,s}} e_{s,k_i} \right) / N_{f,s}$, where $k_i$ is a complementor that provides software $i$. As the value of $v^{1,Embeddedness}_{f,s}$ increases, the degree of embeddedness in the platform also increases. In this situation, the influence of complementors that adopt multi-homing is low.

- Influence of related platforms

To test this, this study includes the variables of influence of related platforms. We calculated the share of software sales of the preceding and succeeding platforms in comparison with platform $f$ as $v^{1,SalesShare,prev}_{f,s}$ and $v^{1,SalesShare,next}_{f,s}$, where $v^{1,SalesShare,prev}_{f,s} = Y_{f^{prev},s} / \left\{ Y_{f^{prev},s} + Y_{f,s} \right\}$, $v^{1,SalesShare,next}_{f,s} =$

$Y_{f^{next},s} / \left\{ Y_{f^{next},s} + Y_{f,s} \right\}$, $Y$ is the total software sales for each condition, $f^{prev}$ was the preceding platform of platform $f$, and $f^{next}$ was the succeeding platform of platform $f$.

### 2.2.3.2. Variables: Environment of Platform Ecosystem

- Bias of sales volume of complementors

We described the degree of sales volume bias as $v_{f,s}^{2,SalesBias}$, and we calculated the value as follows. [a] We calculated the sales volume of each complementor $\acute{Y}_{k,f,s}$, which includes annual software sales provided during time period $s$ for platform $f$. Since the sales volume of software converges to zero in about three months from the release data on average, the sales volume a few months after the release would contain enough of the information required to consider the results of the release. [b] We calculated the HHI of $\acute{Y}_{k,f,s}$ and defined the value as $v_{f,s}^{2,SalesBias}$.

- Bias of scale of the product category of complementary goods

We described the degree of bias in the scale of complementary goods' product category (software genres) as $v_{f,s}^{2,GenreBias}$, and we calculated the value as follows. [a] For each genre $g$, we counted the number of game software $N_{g,f,s}$ provided during time period $s$ for platform $f$. [b] We calculated the HHI of $N_{g,f,s}$ and defined the value as $v_{f,s}^{2,GenreBias}$.

- Growth and decline of the platform ecosystem

The lifecycle of the video game market repeatedly experiences highs and lows. However, the period of each lifecycle stage may not be constant. Additionally, there may be a pattern of renewed growth after a temporary decline. Expressing these unstable patterns using one variable is difficult. For example, the PlayStation Portable platform in Japan has grown again after declining a few years after its release, resulting in the existence of a long-sustained platform [46]. Therefore, we used dummy variables to classify growth and decline of the platform ecosystem. Referring to a previous study on product lifecycles [68], we defined dummy variables for the growth and decline of the platform ecosystem as follows. [a] We calculated the share of game software provided $S_{f,s} \left( = N_{f,s}/N_s \right)$ for each time period $s$ and platform $f$. [b] We calculated the annual variation in the share of game software provided $\Delta S_{f,s} \left( = S_{f,s} - S_{f,s-12} \right)$. [c] We collected values of $\Delta S_{f,s}$ for all cases of $s$ and $f$. [d] We calculated the standard deviation $\sigma$ of the collection of $\Delta S_{f,s}$, and classified the lifecycle stages at time period $s$ on platform $f$ as follows: stage 1 (introduction, $S_{f,s} < 0.05 \times \max \left\{ S_{f,s} \right\}$), stage 2 (growth, $0.5\sigma < \Delta S_{f,s}$), stage 3a (sustained maturity, $0.1\sigma < \Delta S_{f,s} \le 0.5\sigma$), stage 3b (maturity, $-0.1\sigma \le \Delta S_{f,s} \le 0.1\sigma$), stage 3c (declining maturity, $-0.5\sigma \le \Delta S_{f,s} < -0.1\sigma$), and stage 4 (decline, $\Delta S_{f,s} < -0.5\sigma$). [d] For each $s$ and $f$, we set dummy variables for each lifecycle stage, $v_{f,s}^{2,State,1}$, $v_{f,s}^{2,State,2}$, $v_{f,s}^{2,State,3a}$, $v_{f,s}^{2,State,3b}$, $v_{f,s}^{2,State,3c}$, and $v_{f,s}^{2,State,4}$, corresponding to the classification as explained in [d]. We consider this method a more accurate way to extract the lifecycle stage than the previous methods that used time (e.g., [11]).

### 2.2.3.3. Variables: Culture of Complementors in Platform Ecosystem

- Degree of experience in the market

We let the degree of experience in the market be $v_{f,s}^{3,Learning}$. We calculated the value as follows. [a] We counted the number of game software units provided by complementor $k$ over the previous five years until one month prior to the start of time period $s$ (i.e., from [min{$s$} − 60] to [min{$s$} − 1]), and set as $N_{k,s^{p5}}$. In this count, similar types of software were regarded as identical. Here, since complementors that had provided many software units in the past but did not provide any at this time point should be

considered as not having enough experience, we set a five-year period. [b] We calculated $v_{f,s}^{3,Learning}$ as:

$$v_{f,s}^{3,Learning} = \left( \sum_{i}^{N_{f,s}} N_{k_i,s^{p5}} \right) / N_{f,s},$$ where $k_i$ is a complementor that provides software $i$.

- Degree of new participation in the platform

We defined this indicator as follows. [a] We identified two types of complementors in time period $s$ on platform $f$: complementors $k_{f,s}^{mover}$ which have moved from another platform, and complementors $k_{f,s}^{entrance}$ which are new entrants to the video game market. [b] We calculated the rate of provided software by $k_{f,s}^{mover}$ and $k_{f,s}^{entrance}$ and set indicators for the degree of new participation in the platform as $v_{f,s}^{3,Rate,Mover} \left( = N_{k_{f,s}^{mover},f,s} / N_{f,s} \right)$ and $v_{f,s}^{3,Rate,Entrance} \left( = N_{k_{f,s}^{entrance},f,s} / N_{f,s} \right)$.

### 2.2.3.4. Summary of Variables

The variables we considered are summarized in Table 3. This study defined nine variables that could affect the value of influence $\alpha_{f,t}$. Then, we statistically tested how these variables changed influence $\alpha_{f,t}$ using panel-data regression analysis.

**Table 3.** Summary of variables in analysis 2.

| Class | Variable | Expression |
|---|---|---|
| 1. Relationship among platform ecosystems | Degree of monopolization of platforms in the market | $v_{f,s}^{1,MarketHHI}$ |
| 1. Relationship among platform ecosystems | Similarity of product category of complementary goods | $v_{f,s}^{1,Sim.Comp.}$ |
| 1. Relationship among platform ecosystems | Embeddedness of complementors in the platforms | $v_{f,s}^{1,Embeddedness}$ |
| 1. Relationship among platform ecosystems | Influence from related platforms: preceding one and succeeding one | $v_{f,s}^{1,SalesShare,prev}, v_{f,s}^{1,SalesShare,next}$ |
| 2. Environment of the platform ecosystem | Bias of sales volume of complementors | $v_{f,s}^{2,SalesBias}$ |
| 2. Environment of the platform ecosystem | Bias of scale of product category of complementary goods | $v_{f,s}^{2,GenreBias}$ |
| 2. Environment of the platform ecosystem | Growth and decline of the platform ecosystem: lifecycle stage 1 (introduction), stage 2 (growth), stage 3a (sustained maturity), stage 3b (maturity), stage 3c (declining maturity), and stage 4 (decline) | $v_{f,s}^{2,State,1}, v_{f,s}^{2,State,2}, v_{f,s}^{2,State,3a}, v_{f,s}^{2,State,3b}, v_{f,s}^{2,State,3c}, v_{f,s}^{2,State,4}$ |
| 3. Culture of complementors in the platform ecosystem | Degree of experience in the market | $v_{f,s}^{3,Experience}$ |
| 3. Culture of complementors in the platform ecosystem | Degree of new participation in the platform: rate of movers from other platforms, and rate of entrants in the market | $v_{f,s}^{3,Rate,Mover}, v_{f,s}^{3,Rate,Entrance}$ |

### 2.2.3.5. Statistical Method

We set $\alpha_{f,t}$ as the dependent variable and set the variables outlined in Sections 2.2.3.1 to 2.2.3.3 as independent variables. We defined time period $s$ as a moving window of 12 months. The settings for time period $s$ and $t$ correspond to those in analysis 1 (i.e., $s = t - 11, \ldots, t$). This 12-month window removes the seasonality bias and mitigates the influence of killer software.

Since our dataset comprises panel data, we undertook a panel-data analysis in which we constructed four models. Model 1 included variables of relationship among platform ecosystems. Since most previous studies about platform competition have discussed these variables, we set this model as our base model. Model 2-a added variables for the environment of the platform ecosystem to

model 1, while model 2-b added variables for the culture of complementors in the platform ecosystem to model 1. Model 3 was the full model and included all the independent variables. To select a proper method for the panel-data analysis, we tested three methods (pooling model, fixed-effects model, and random-effects model) for each model with the dataset using an F-test (used to compare between the pooling model and the fixed-effects model), a Breusch–Pagan test (to compare between the pooling model and the random-effects model) and a Hausman test (to compare between the fixed-effects model and the random-effects model). Once we confirmed that the random-effects model was the most appropriate for all our models, we adopted it for our analysis. Accordingly, model 3 is described as:

$$
\begin{aligned}
\alpha_{f,t} = \beta_1 v_{f,s}^{1,MarketHHI} &+ \beta_2 v_{f,s}^{1,Sim.Comp.} + \beta_3 v_{f,s}^{1,Embeddedness} + \beta_4 v_{f,s}^{1,SalesShare,prev} \\
&+ \beta_5 v_{f,s}^{1,SalesShare,next} + \beta_6 v_{f,s}^{2,SalesBias} + \beta_7 v_{f,s}^{2,GenreBias} + \beta_8 v_{f,s}^{2,State,1} \\
&+ \beta_9 v_{f,s}^{2,State,2} + \beta_{10} v_{f,s}^{2,State,3a} + \beta_{11} v_{f,s}^{2,State,3c} + \beta_{12} v_{f,s}^{2,State,4} \\
&+ \beta_{13} v_{f,s}^{3,Experience} + \beta_{14} v_{f,s}^{3,Rate,Mover} + \beta_{15} v_{f,s}^{3,Rate,Entrance} + \mu_f + \varepsilon_{f,s,}
\end{aligned}
\tag{6}
$$

where, each $\beta$ was the coefficient for each variable, $\mu_f$ was the random effect for platform $f$, and $\varepsilon_{f,t}$ was an error term. Here, the variable for maturity stage $v_{f,t}^{2,State,3b}$ was not included, since we set that variable as the standard for the variables of the growth and decline of the platform ecosystem.

Next, to ensure statistical validity, we did the following. First, we removed samples, including outliers, related to the decline of platform ecosystems. In the video game market, platforms usually continue with low market shares even after they decline, and they might not die completely (mainly because of the provision of cross-platform software that is compatible with or has the same content as new platforms). Since such periods could cause a bias in the distribution of the dataset, we removed those samples, including two conditions, as follows: (a) $N_{f,s} < 0.05 \times \max\left(N_{f,s}\right)$ is satisfied in time period $s$, (b) and $N_{f,s^{after}} < 0.05 \times \max\left(N_{f,s^{after}}\right)$ is satisfied in any time period $s^{after}$ after time period $s$.

Second, we restricted the samples according to the nature of the dependent variable $\alpha_{ft}$ to ensure the normal distribution of the error term. Although we conducted premodification in Section 2.2.2.2, we still found that samples that included large values of $\left|\alpha_{ft}\right|$ could lose the normal distribution of the error term. Through the pre-repetition test, we confirmed that the condition of $\left|\alpha_{ft}\right| \leq 0.23$ was appropriate for restricting the samples. In this case, a normal distribution of error terms for the full model was ensured ($p > 0.05$ with Shapiro–Wilk test), although the sample was reduced to 68 percent of its initial size.

Third, we dealt with the heterogeneity of variance and serial correlation depending on the nature of the panel data by calculating the clustered robust standard error and modified errors as well as the statistical significance of each independent variable.

Finally, we confirmed that there was no multicollinearity in any of our models (mean VIF = 2.00 in full model).

## 3. Results

### 3.1. Results of Analysis 1

Figure 6 shows the results of analysis 1, on the distribution of the extracted values of influence $\alpha_{f,t}$. The x-axis represents the value of $\alpha_{f,t}$, and the y-axis represents the number of times $\alpha_{f,t}$ occurs in units of 0.1. As $\alpha_{f,t}$ approaches $-1.0$, complementors on platform $f$ receive a stronger symbiotic influence via the growth and decline of competitor platform ecosystems. Conversely, as $\alpha_{f,t}$ approaches 1.0, complementors on platform $f$ receive a stronger competitive influence via the growth and decline of competitor platform ecosystems.

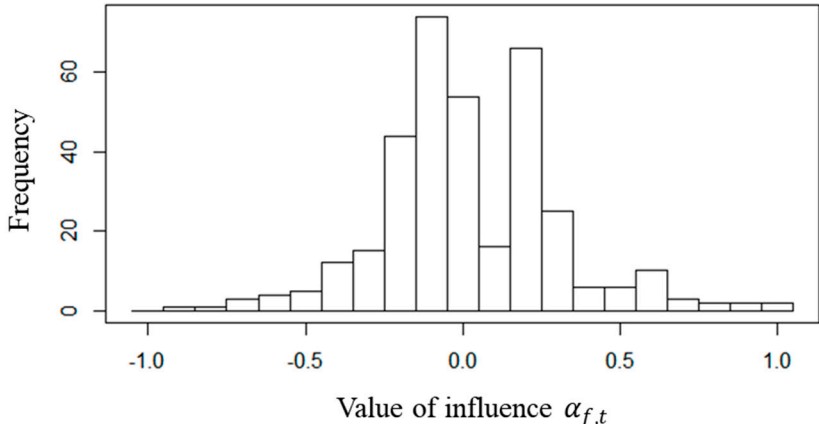

**Figure 6.** Results of analysis 1: distribution of values of influence $\alpha_{f,t}$. As $\alpha_{f,t}$ increases, it approaches a winner-takes-all situation. As $\alpha_{f,t}$ decreases, it approaches symbiosis.

The results indicate that the value of $\alpha_{f,t}$ in the video game industry could be either positive or negative. In the distribution, the most frequent value is about $-0.1$ within the negative area and about $0.2$ within the positive area. Accordingly, our results indicate that complementors on platform $f$ usually receive an influence of about 10 to 20 percent from the growth and decline of competitor platform ecosystems. Additionally, the rate of samples for $\alpha_{f,t} < 0$ was about 45 percent, although the rate of the sample numbers for $\alpha_{f,t} > 0$ was about 39 percent. Accordingly, the values of influence $\alpha_{f,t}$ related to symbiotic situations appeared equal to or more than the values of influence $\alpha_{f,t}$ reflecting winner-takes-all situations. From these results, we can confirm that complementors act more frequently to bring about symbiotic situations for platform ecosystems than they do to produce winner-takes-all situations in the Japanese video game market.

### 3.2. Results of Analysis 2

Table 4 shows the regression results of analysis 2. As mentioned, model 1 is our base model, model 3 is the full model, and models 2-a and 2-b are the intermediate models. In this table, the dummy variable for the introduction stage $v_{f,t}^{2,State,1}$ is not included, since no sample satisfies the introduction stage classification criteria in our data. The dummy variable for maturity stage $v_{f,t}^{2,State,3b}$ is also not included, since we set this variable as the standard for the variables of the growth and decline of the platform ecosystem. In the regression calculation, the variables (excluding dummies) are standardized as the z-score (i.e., mean value is 0 and standard deviation is 1).

The adjusted $R^2$ coefficient of determination for the full model is 0.64, which is higher than that of the base model (about 0.36). Accordingly, our results indicate that the addition of the variables reflecting the environment of the platform ecosystem and complementor culture significantly improves the value of influence $\alpha_{f,t}$.

Regarding the indicators of the relationship among platform ecosystems, the embeddedness of complementors in the platforms, $v_{f,t}^{1,Embeddedness}$, and the influence of the related preceding platform, $v_{f,t}^{1,SalesShare,prev}$, have significant negative effects on $\alpha_{f,t}$ (both $p < 0.01$). Thus, the results indicate that complementors tend to act symbiotically in platform $f$ as these two variables increase. Accordingly, hypothesis 1 was partially supported.

Regarding the indicators of the platform ecosystem environment, the bias of complementors' sales volume, $v_{f,t}^{2,SalesBias}$, has significantly positive effects on $\alpha_{f,t}$ ($p < 0.05$), and the bias in the scale of complementary goods' product category, $v_{f,t}^{2,GenreBias}$, has significantly negative effects on $\alpha_{f,t}$ ($p < 0.05$). Accordingly, the results indicate that an increased bias in software sales facilitates a winner-takes-all competition in platform $f$, while an increase in the of bias in software genres facilitates a symbiotic state. In addition, regarding the indicators of the growth and decline of the platform ecosystem,

stage of growth, $v_{f,t}^{2,State,2}$, has significantly negative effects on $\alpha_{f,t}$ ($p < 0.01$), and stage of decline, $v_{f,t}^{2,State,4}$, has significantly positive effects on $\alpha_{f,t}$ ($p < 0.01$). Accordingly, the results indicate that an ecosystem's decline facilitates a competitive state between platform $f$ and other platforms, but an ecosystem's growth facilitates a symbiotic state between platform $f$ and the complementors. Accordingly, hypothesis 2 was partially supported.

In case of indicators of culture of complementors in the platform ecosystem, the degree of experience in the market $v_{f,t}^{3,Experience}$ has significantly negative effects on $\alpha_{f,t}$ ($p < 0.05$). Accordingly, the results indicate that an increase in the degree of experience in the market drives the symbiotic states between platform $f$ and other platforms via the complementors. In addition, as the degree of new participation in the platform increases, the rate of movers from other platforms, $v_{f,t}^{3,Rate,Mover}$, has significantly negative effects on $\alpha_{f,t}$ ($p < 0.01$), and the rate of entrants to the market, $v_{f,t}^{3,Rate,Entrance}$, has significantly positive effects on $\alpha_{f,t}$ ($p < 0.01$). Accordingly, the results indicate that an increase in the rate of movers from other platforms drives a symbiotic state between platform $f$ and other platforms, whereas an increase in the rate of entrants in the market creates a winner-takes-all competition in platform $f$. Accordingly, hypothesis 3 was supported.

**Table 4.** Results of analysis 2: how influence $\alpha_{f,t}$ is affected by the relationship among platform ecosystems, the environment, and the culture of complementors.

| Variable | Model 1 | | Model 2-a | | Model 2-b | | Model 3 | |
|---|---|---|---|---|---|---|---|---|
| Monopolization of the market | 0.06 | ** | 0.01 | | 0.05 | ** | 0.00 | |
| | (0.01) | | (0.01) | | (0.01) | | (0.01) | |
| Similarity of product category | −0.01 | | 0.01 | | −0.02 | | 0.01 | |
| | (0.03) | | (0.01) | | (0.02) | | (0.01) | |
| Embeddedness of complementors | −0.08 | ** | −0.06 | ** | −0.13 | ** | −0.07 | ** |
| | (0.03) | | (0.01) | | (0.04) | | (0.02) | |
| Influence of preceding platform | −0.03 | ** | −0.02 | ** | −0.03 | ** | −0.02 | ** |
| | (0.01) | | (0.00) | | (0.01) | | (0.01) | |
| Influence of succeeding platform | 0.01 | | 0.00 | | 0.00 | | 0.00 | |
| | (0.01) | | (0.00) | | (0.01) | | (0.00) | |
| Bias of sales volume | | | 0.05 | ** | | | 0.04 | * |
| | | | (0.01) | | | | (0.02) | |
| Bias of product category | | | −0.02 | | | | −0.03 | * |
| | | | (0.02) | | | | (0.01) | |
| Lifecycle stage: growth | | | −0.11 | * | | | −0.11 | ** |
| | | | (0.05) | | | | (0.03) | |
| Lifecycle stage: sustained maturity | | | −0.05 | | | | −0.05 | |
| | | | (0.04) | | | | (0.03) | |
| Lifecycle stage: declining maturity | | | 0.03 | * | | | 0.01 | |
| | | | (0.01) | | | | (0.01) | |
| Lifecycle stage: decline | | | 0.10 | ** | | | 0.09 | ** |
| | | | (0.03) | | | | (0.03) | |
| Experience in the market | | | | | −0.03 | * | −0.03 | * |
| | | | | | (0.02) | | (0.01) | |
| Rate of movers from other platforms | | | | | −0.06 | ** | −0.03 | ** |
| | | | | | (0.01) | | (0.01) | |
| Rate of entrants to the market | | | | | 0.00 | | 0.03 | ** |
| | | | | | (0.01) | | (0.01) | |
| Intercept | 0.02 | | 0.03 | | 0.03 | | 0.03 | |
| | (0.04) | | (0.04) | | (0.04) | | (0.03) | |
| Adjusted $R^2$ | 0.28 | | 0.60 | | 0.45 | | 0.64 | |

Note: ** = $p < 0.01$; * = $p < 0.05$.

## 4. Discussion

### 4.1. Interpretation of Results

This study had two purposes: (a) to confirm that complementors usually act to avoid a winner-takes-all situation (i.e., to induce a symbiotic situation), and (b) to comprehensively clarify the underlying mechanisms of complementors' behavior which brings winner-takes-all (or symbiotic) situations. We analyzed influence $\alpha_{f,t}$ as the degree to which complementors are affected by the performance of competing platform ecosystems. Analysis 1 showed that complementors act more frequently to bring about symbiosis for platform ecosystems than they do to produce winner-takes-all situations. Analysis 2 showed that the value of influence $\alpha_{f,t}$ changes due to factors like the relationship among platform ecosystems, the environment, and the culture of complementors in the ecosystem.

Analysis 2 indicates that first, among the factors reflecting the relationships among platform ecosystems, an increase in Embeddedness of complementors and Influence of preceding platform were the influential for avoiding a winner-takes-all situation. A high value for complementor embeddedness indicates that each complementor invest in fewer platforms. This result is likely due to the gap in profitability among platforms, the difficulty of cross-platforms, or the ease of participation enjoyed by small-scale firms. Although a previous study has suggested that a platform architecture that facilitates cross-platforms decreases the possibility of winner-takes-all situations [40], our results imply that the ease of cross-platforms instead increases the chances of a winner-takes-all competition. Regarding the indicator for the influence of preceding platforms, a high value indicates that the platform has a large pool of complementors that can move to the new platform. Accordingly, this shows that these complementors will switch to the new platform when a new generation of platform ecosystems starts to emerge.

Second, among the platform ecosystem environment factors, we confirmed that an increase in Bias of sales volume and Lifecycle stage: decline would lead to a winner-takes-all situation, while Bias of product category and Lifecycle stage: growth would lead to the avoidance of a winner-takes-all situation. A high value of bias of sales volume reflects intensified competition and a larger sales gap among the complementors in the platform ecosystem. This result indicates that complementors that earn low profits switch to more profitable platforms. Previous studies have suggested that complementors earning low profits will either exit the platform or move to other platforms [3,46], and our analysis supports this view. Regarding product category bias, the result shows that a high value induces complementors to stay on the platform, although it could lead to intensified competition. A previous study has suggested that differentiated platform positioning weakens winner-takes-all [14] situations, and our results support this view. Regarding the lifecycle stage factors, the growth and decline stages had contrary effects on the actions of complementors. This likely indicates that complementors imitate other complementors, such as in the bandwagon effect [49–51]. We found that an increase in investment by complementors reduces winner-takes-all competition among the platforms, whereas a decrease in investment by complementors induces winner-takes-all competition; the conclusion of the competition would be decided faster if the investment reduction and winner-takes-all situation occurred simultaneously. We also conclude that complementors' imitative actions induce the alternation between winner-takes-all and symbiosis situations, as shown in Figure 1.

Finally, regarding the factors reflecting the culture of the complementors in the platform ecosystem, we confirmed that an increase in Rate of entrants to the market leads to a winner-takes-all situation, whereas increases in Experience in the market and Rate of movers from other platforms lead to an avoidance of winner-takes-all situations. A high value of Experience in the market indicates that there are many complementors with wide experience in new product development. Since these complementors would be embedded in the market, they could maintain not only certain platform ecosystems but also the existing scale of the market. We consider that since newcomers would not have such motivations (maintaining the entire market), they would tend to act in ways that lead

to a winner-takes-all situation. The results showing a high rate of entrants to the market would be consistent with this view.

### 4.2. Theoretical Implications

This study contributes to the literature by identifying the mechanisms behind winner-takes-all competition. The nature of platform ecosystems is such that indirect network effects occur between their complementors and consumers [1,2]. Previous studies have suggested that this effect could cause a winner-takes-all situation, whereby a single platform takes most of the complementors and consumers [27–29]. This study has shown that platform ecosystems have not only winner-takes-all competitions but also symbiotic ones. These are induced by actions taken by complementors for goods provision, and they are influenced by several platform ecosystem factors.

Our results provide implications for future research on ecosystem in the management field. Scholars have suggested that keystone firms are significant in a business ecosystem [69]. The concept of "keystone firms" was borrowed from the literature on biological evolution. Keystone firms serve as hubs in networks of ecosystem interactions and can improve overall chances of survival in the face of change by providing benefits to the ecosystem as a whole [69]. Scholars argue that platforms and platform providers could become keystone firms [31]. However, our study suggests that certain complementors could also contribute to a symbiotic survival and coevolution among platform ecosystems by functioning as keystone firms.

When competition ends in winner-takes-all, the market has been monopolized by a single platform and provider. In such a situation, the platform provider has a free hand to maximize its profits, which would demand more profit allocation from both the complementors and the consumers. This would cause the complementors and consumers to become unsustainable within the ecosystem and in turn reduce the stability and survivability of the platform ecosystem itself. The complementors, which provide goods for multiple platform ecosystems and behave in symbiotic ways, serve as keystone firms in the platform ecosystems by keeping the competition moderate. Our results indicate that complementors with more experience in the market could become such keystone firms. Additionally, certain environmental factors of the platform ecosystem, which is distinct from other platform ecosystems and ensures the profitability of complementors, could help the complementors become keystone firms. Thus, our study suggests the emergence of new types of keystone firms in platform ecosystems. These keystone complementors could improve the sustainability of platform-based markets and facilitate the coexistence of multiple platform ecosystems.

### 4.3. Managerial Implications

This study also provides implications for platform providers concerning the management of their ecosystems. We suggest that platform providers could manage the influence of competing platforms for complementary goods provision by managing the indicators in the ecosystem. We suggest that adjusting the platform relationships according to the situation is important for fostering platform ecosystem growth. This study also provides important implications for governments seeking to grow their national markets. If the relationship among platform ecosystems in the market is competitive, market growth will be restricted, as Equation (4) shows. If the relationship is symbiotic, however, the market will grow. Thus, the results imply that governments should promote conditions that are favorable for symbiosis in order to develop platform-based markets.

### 4.4. Limitations and Future Research

This study has several limitations. First, it does not consider the distinctiveness of individual platforms, such as their technological features or the strategies of the platform providers. Although this study controlled for the influences arising from platform distinctiveness via a panel-data analysis, the inclusion of platform distinctiveness could enable a deeper understanding of the mechanisms behind the relationships among platform ecosystems. Nintendo Wii is an example of a video game

platform that has a distinctive technology and market strategy. Nintendo Wii adapted its strategy of cultivating new markets comprised of casual users and developed a unique video game system in an environment of closed innovation to achieve that goal [3,70–72]. Technologies and platform market strategies that are similarly distinct might influence the goods provision of complementors and new product development.

Second, the results of this study are affected by the structure of the video game market, which consists of both hardware and software. In such a structure, complementors develop their goods by using platform technologies. However, a platform-based market encompasses not only such high-technology industries but also more wide-ranging settings, such as shopping malls, stock exchanges, single-serve coffee makers, real estate brokerages, and health maintenance organizations [73]. Service intermediation platforms do not have a hardware–software structure. Several studies have focused on service intermediation platforms, such as online travel agents [74], hotel reservation services [75], taxi–hiring services [76], and grocery-delivery services in the sharing economy [77]. These platforms usually act as an intermediary between existing services rather than develop new services. The game industry also includes mobile game platforms (e.g., App Store and Google Play, and smartphone devices) and web game platforms (e.g., Steam), in addition to the video game platforms this study analyzed. These other game platforms have differences from video game platforms, such as lengths of playtime for each game, business model pricing schemes, and degrees of structural differences among the platforms. Accordingly, the decision-making mechanisms of complementors on these platforms could be very different from the mechanisms of those on hardware–software platforms. Our concept regarding the competition and symbiosis mechanisms among platform ecosystems could be adapted to any platform ecosystem. However, the probability of the occurrence of each situation and the degree to which each influential factor may change a situation will depend on the features of the platform and the market structure of the ecosystem. Hence, future studies should focus on other types of platform to analyze the relationships among platform ecosystems.

Third, this study did not analyze the consumer side. Consumers in video game markets, unlike complementors, do not usually require additional actions for the platform once it has been installed (excluding the adoption of expansions). Accordingly, we assumed that consumers may not be invested in either competition or symbiosis. However, it would be worthwhile investigating the influence on consumers of the actions of complementors amid either competition or symbiosis. In addition, since consumers could incur direct network effects through platform functions such as online reviews (e.g., [78]), these features on the consumer side might influence platform relationships. Thus, future studies should focus on the consumer side to analyze the relationships among platform ecosystems.

Fourth, this study serves as the first step in defining and analyzing competition and symbiosis at equivalent levels for platform ecosystems. We therefore verified the existence of states of symbiosis and examined how several factors affected the occurrence of competitive and symbiotic situations. This study investigated broad explanatory variables and did not verify moderation effects between variables and potential boundary conditions. Future research could explore such moderation effects and potential boundary conditions from among the influence factors identified in this study.

Finally, this study is the first to clarify the mechanisms of symbiosis in platform ecosystems. We consider that symbiotic situations among platform ecosystems will enhance the sustainability of ecosystems and their platform-based markets. However, the achievement of this study is limited to the extraction of competitive/symbiotic situations and an analysis of the factors influencing these situations. Future research could clarify how a competitive/symbiotic situation affects the sustainability of platform ecosystems.

**Funding:** This work was funded by JSPS KAKENHI Grant Numbers 18K12874.

**Acknowledgments:** We thank Masaharu Tsujimoto, Takeshi Takenaka, and Hiroki Takahashi for their many helpful comments.

**Conflicts of Interest:** The author declares no conflict of interest.

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
