# Peer review of "Winner-Takes-All or Co-Evolution among Platform Ecosystems: A Look at the Competitive and Symbiotic Actions of Complementors"

_sustainability, doi:10.3390/su11030726_

Reviewer 1 Report

The research paper adapts a model of ecosystem dynamics stemming from the biology and applies it in the video-game industry business “ecosystem”. The researchers study the factors, and conditions that lead the competition among “ecosystems” towards “winner-takes-all” instances or a symbiotic relationship. Using a dataset from the Japanese video games industry, they empirically estimate the parameters of their model. This piece of research is original, and raises some very interesting and important issues vis-à-vis the video games industry and the organisational ecology theory. However, there are some major issues that prevents the research from hitting its mark that can be summarised into two main types: a) very weak theoretical grounding which prevents the reader to assess the structural validity of the research and b) the structure of the paper. I hope my comments will help the authors to improve the clarity of the manuscript.

Major issues:

1.      The problem begins in the abstract. It has significant number of jargon that does not allow the reader to understand the scope of the research. The abstract needs to be crisp clear and explain exactly the unit of analysis, method and findings. Is it a single ecosystem (18: complementors among ecosystems) or between ecosystems?

2.      The authors should define early on the “ecosystem” in the business context. In biology, they are considered closed systems, and as such equilibria can be achieved. In the business world this is rarely the case and as a consequence this has significant implications that are to be raised and discussed in the paper (line 220).

3.      Throughout the paper the authors use structures such as: “ecosystem” “platform” and “two-sided markets”. If those structures refer to a single one, please be consistent with the terminology. Otherwise provide definitions of those to understand the differences. I,e in 52-53 lines

4.      Explain what the “symbiosis” in business context is. Is there literature about it?

5.      The paper does not take into consideration in the sample the mobile games market. That is a limitation that needs to be acknowledged and discussed.

6.      Apparently, the authors use a typology of actors in those ecosystems but we are not informed about the types and the dynamics between those actors and the ecosystems. This is important in order to improve the positioning of the paper in the current literature.

7.      142: The authors introduce the research questions. However, up to that point, there has been no meaningful discussion of the current literature, a critical review of that, and how the literature informed these research questions. As a result, the questions feel meaningless and without context. There is a need for a coherent section where the authors discuss comprehensively the current literature that informed their research.

8.      As on comment 7, the framework of analysis does not provide any meaningful insight to the reader as there is not a discussion about the model, how it was used in the past and its limitations. Instead, it appears later into the discussion but up to that point it raises many questions and doubts. This describes one of the main issues of the paper. Due to structural limitations, the authors introduce constantly terms, structures, variables without enough discussion or justification. This can be solved through a thorough overhaul of the paper’s structure which I strongly encourage the readers to do.

9.      160: variables such as carrying capacity is important for the analysis. However, they remain a biological term and construct. Is there an equivalent in the business literature? What are the differences. This raises another important issue. Without a meaningful theoretical review of the paper I cannot comment on whether the structures chosen by the authors are valid or not. This casts shadows to the paper’s structural validity.

10.  424. Minimal discussion of organisational learning, and how the measure captures organisational learning sufficiently. Please expand.

Minor issues

1.      40: The success of the platform ecosystem depends on the success of the ecosystems. Please clarify. Too vague.

2.      75: Explain the HHI index and justify its choice

3.      77: Please explain the market concentration is it C4 or is C7?

4.      91-92 The authors argue that technological superiority of the platform (I think they mean the console) would allow the creation of an ecosystem. Historical evidence show that technological superiority does not have necessarily that effect because of the potential network effects and lock in effects of the player base. Please clarify.

5.      122 How can a platform be independent of another platform in a single industry? Please clarify

6.      137: how the profitability of other ecosystems influences the activities of complementors. Please clarify.

7.      160: please define the variables as they are introduced: Species, carrying capacity.

8.      179: move the paragraph’s discussion earlier, preferably in a literature review related section.

9.      188: measures need to be clearer and crisply described.

10.  210: move to a literature review section

11.  232. Please provide information about sample size, justification of dataset, types of variables, steps of sample selection, impact on size, etc. Discuss and evaluate bias stemming from the sample selection as it is used to estimate the model’s parameters.

12.  312: the variable summary needs to appear in the methodology or ample section.

I hope my comments and suggestions will encourage the authors to improve the manuscript and highlight better their contribution to knowledge.

Author Response

We wish to express our appreciation to the Reviewer for their insightful comments on our paper. The comments have helped us significantly improve the paper. We attended to all proposed comments and revised our manuscript, as the attached file.

Reviewer 2 Report

Dear Author(s),

Glad to review your paper "Winner-takes-all or co-evolution among platform ecosystems: A look at the competitive and symbiotic actions of complementors". 

The paper is well structured, scientifically robust and it investigates an interesting topic.

However, I would recommend you to consider:

- the work starts from the difference between technological platforms and platform ecosystems which is considered as given. I think you could elaborate on the differences between the two concepts also to avoid mistakenly overlaps

- You may want to clarify the difference between platform as a market and platform as technology drivers respectively of competition and innovation. That doesn't come through your argument and they seem to be adopted interchangeably.

- Not clear enough the "why" of your research. What did push you to investigate this topic? It looks more like a pure theoretical argument rather than a research driven or inspired by a practical issue. I understand - and I personally like - theoretical contributions. However, it looks like here there could be some but they haven't be made explicit.

Good luck and kind regards,

Author Response

(The authors gave the same response as above.)

Reviewer 3 Report

The paper addresses a very important angle that has not been adequately addressed in prior research. Indeed, symbiotic situations are quite relevant while prior platform research is mostly focused on winner-takes-all mentality. The paper is inspired by dynamics of biological systems which is interesting and worthwhile. Relevant variables have been used and operationalization are clear. I have few suggestions for improvement:

1- The paper can benefit from formal hypotheses development. At this stage, the reason behind the relationships are not clearly and theoretically discussed. 

2- Variable life cycle seems to be ordinal and the dummy variables indicates moving from negative to positive. It is not clear why the authors have used dummy variable instead of one variable for life stage.

3- No moderation effect is tested and potential boundary conditions and interactions are not discussed. This makes the model a bit simplistic.

4- Discussion section can also benefit form considering boundary conditions to the findings, for example whether the findings are applicable to other platform ecosystems such as industry platforms.

5- Consideration of multiplicity of ecosystems is rarely studied and it is an advantage of this paper, However it is good to include few references that have looked into such situations, for example SMJ piece by Pierce, (2009).

I wish you best of luck in finalizing the paper.

Reference

Pierce L. 2009. Big losses in ecosystem niches: how core firm decisions drive complementary product shakeouts. Strategic Management Journal 30(3): 323–347.

Author Response

We wish to express our appreciation to the Reviewer for their insightful comments on our paper. The comments have helped us significantly improve the paper. We attended to all proposed comments and revised our manuscript, as the attached file.

Round  2

Reviewer 1 Report

I suggest some minor improvements before the publication of the paper.

C4 concentration refers to the market concentration of the four biggest companies of the industry as opposed to C7 which refers to the 7 biggest; not to profits and market share respectively

Minor typos and spell checks need to be addressed before publication. Given the limited time provided to the authors, it is understandable but I suggest a thorough proofreading of the manuscript.

409: “the total number of samples is 1409 […]” do the authors refer to the size of the sample?

Hypotheses are expressed in technical terms. Simplification may be required to make allow a broader audience for the paper. (for instance, instead of referring to α, the authors could explicitly mention what α stands for)

Author Response

Dear Reviewer 1,

We wish to express our gratitude for your insightful comments on our paper. The comments have helped us significantly improve the paper. We attended to all proposed comments and revised our manuscript, as follows.

[Comment 1]                                                     

C4 concentration refers to the market concentration of the four biggest companies of the industry as opposed to C7 which refers to the 7 biggest; not to profits and market share respectively.

[Response 1]

We appreciate the further explanation of C4 and C7.

This market concentration, which is expressed by the HHI (Herfindahl–Hirschman Index), reflects how much of a market a company occupies. Accordingly, market concentration does not reflect the concentration of the four or seven biggest companies. The value of HHI can reflect how many companies occupy the market.

[Comment 2]

Minor typos and spell checks need to be addressed before publication. Given the limited time provided to the authors, it is understandable but I suggest a thorough proofreading of the manuscript.

[Response 2]

In accordance with the Reviewer’s comment, we submitted the manuscript to an English editing service for overall language checks, including for typos and spelling.

[Comment 3]

409: “the total number of samples is 1,240 […]” do the authors refer to the size of the sample?

[Response 3]

As you suggested, “the total number of samples” means “the size of the sample.” We revised the description to clarify as follows:

“The sample comprised 1,240 observations.”

(at line 410)

[Comment 4]

Hypotheses are expressed in technical terms. Simplification may be required to make allow a broader audience for the paper. (for instance, instead of referring to α, the authors could explicitly mention what α stands for)

[Response 4]

In accordance with the Reviewer’s comment, we revised the description of the hypotheses. Since we consider that referring to α in the hypotheses is necessary, we added a supplementary description of α instead of removing α. As adding further descriptions to explain the other technical terms would make the hypothesis descriptions too long, we added only the explanation of α. We revised as follows:

Hypothesis 1: The value of influence α (the degree to which complementors are affected by the performance of competing platform ecosystems) is affected by four factors related to the relationships among platform ecosystems: Degree of monopolization of platforms in the market, Similarity of product category of complementary goods, Embeddedness of complementors in the platforms, and Influence of related platforms.

We similarly revised each hypothesis:

(at line 236)

(at line 266)

(at line 292)

Reviewer 3 Report

Thanks for the efforts in developing the theoretical contributions and hypothesis section. I think it is now a much better revision.

Author Response

Dear Reviewer 3,

We again wish to express our gratitude for your comments on our paper.

Since you indicated that “English language and style are fine/minor spell check required,” we submitted the manuscript to an English editing service for overall language checks.